# High-low pressure domain wall for the classical Toda lattice

Christian B. Mendl[1*] and Herbert Spohn[2†]

**1** Department of Informatics and Institute for Advanced Study,
Technical University of Munich, Boltzmannstraße 3, 85748 Garching, Germany
**2** Department of Mathematics and Department of Physics, Technical University of Munich,
Boltzmannstraße 3, 85748 Garching, Germany

⋆ christian.mendl@tum.de, † spohn@tum.de

## Abstract

We study the classical Toda lattice with domain wall initial conditions, for which left and right half lattice are in thermal equilibrium but with distinct parameters of pressure, mean velocity, and temperature. In the hydrodynamic regime the respective space-time profiles scale ballisticly. The particular case of interest is a jump from low to high pressure at uniform temperature and zero mean velocity. Thereby the scaling function for the average stretch (also free volume) is forced to change sign. By direct inspection, the hydrodynamic equations for the Toda lattice seem to be singular at zero stretch. In our contribution we report on numerical solutions and convincingly establish that nevertheless the self-similar solution exhibits smooth behavior.


# 1 Introduction

Over the past decade the out-of-equilibrium dynamics of many-body systems in one dimension has received a lot of attention, see for example the recent survey article [1]. Besides the laws governing the dynamics, also the initial conditions have to be specified. One popular choice is quantum quench [2, 3], for which one prepares the ground (or thermal) state for a particular hamiltonian and then evolves in time according to some other translation invariant hamiltonian. A further much studied choice, the one discussed in our contribution, is domain wall: In the left/right half lattice one prepares thermal states of a given translation invariant hamiltonian. The product coupling of these states evolves then under the full hamiltonian. If in either half the thermodynamic parameters are identical, one deals with a small perturbation of a global equilibrium state. But, if the parameters differ, the time evolution is far from equilibrium. Besides being a physically natural choice, macroscopic properties can be predicted on the basis of a suitable hydrodynamic theory.

In our contribution we focus on the classical Toda lattice with initial domain wall. The dynamics is integrable and, in principle, generalized hydrodynamics (GHD) can be used to quantitatively describe the ballistic spreading [4, 5]. The average current of GHD is of the form $\nu^{-1}(\nu^{\mathrm{eff}} - q_1)$, where the effective velocity, $\nu^{\mathrm{eff}}$, is the solution of a rate equation and $q_1$ is the average momentum [18]. $\nu$ denotes the average stretch (also free volume), which is defined through $\nu = \langle q_{j+1} - q_j \rangle$ with $q_j$ the position of the $j$-th Toda particle. In thermal equilibrium $\nu$ is independent of $j$. For low pressure the stretch is positive, $\nu > 0$, since particles are far apart. The stretch decreases with increasing $P$ and at some critical pressure, $P_c$, the stretch vanishes, while $\nu < 0$ for $P > P_c$, compare with Fig. 1(d) below. In the latter regime the particles are typically anti-ordered, i.e. $q_j > q_{j+1}$, whereas for $\nu > 0$ they are typically ordered.

The low-high pressure domain wall refers to imposing a pressure $P_- < P_c$ in the left half lattice and $P_+ > P_c$ in the right half, with uniform temperature and zero mean velocity throughout. As a consequence $\nu = \nu_- > 0$ in the left half lattice and $\nu = \nu_+ < 0$ in the right one. On the basis of the hydrodynamic equations, the stretch is expected to reach a self-similar form as $\nu(x,t) = g(x/t)$ with some suitable scaling function $g$. Thus $g(-\infty) = \nu_-$, $g(\infty) = \nu_+$ and the scaling function is forced to vanish at some intermediate spatial location, $g(x_c) = 0$, and hence $\nu(x_c t, t) = 0$. But at this location the average current seems to diverge because of the prefactor $\nu^{-1}$. Thus a more detailed investigation of the self-similar solution is required, which is the topic of the present work. The case of a low-low pressure jump has been studied before [20], including low order quantum corrections. In this case $g$ stays strictly positive and varies smoothly, as anticipated. Of course, a corresponding behavior is expected for a high-high pressure domain wall.

Already in the pioneering contributions [4, 5] it was realized that the solution of the GHD domain wall problem has a very specific structure. In the position-(spectral parameter) plane one has to determine a contact line at which the spectral parameter jumps from its left to its right value. This contact line fully characterizes the solution, which brings us to a second motivation for our study. While the entire self-similar solution of generalized hydrodynamics is computed numerically, most commonly only the spatial dependence of density, momentum, and temperature are displayed. Rather than such low order moments, we regard the contact line as more informative and will numerically determine its shape.

To give an outline: We briefly review the domain wall solution for non-integrable chains. Generalized hydrodynamics of the Toda lattice is recalled. Our main result are numerical solutions of GHD for the low-high pressure domain wall. As a by-product we elucidate the structure of the density of states for the Lax matrix as $P$ varies through $P_c$.

## 2 Domain wall for non-integrable chains

We briefly recall the domain wall problem of conventional hydrodynamics, so to provide a contrast with integrable chains. We start by considering a wave field over $\mathbb{Z}$, displacements denoted by $\{q_j \in \mathbb{R}, j \in \mathbb{Z}\}$, which is governed by

$$\ddot{q}_j(t) = U'(q_{j+1}(t) - q_j(t)) - U'(q_j(t) - q_{j-1}(t)). \tag{1}$$

Here $U$ is a smooth function, bounded from below, with at least a one-sided linear growth at infinity. Note that the right side depends only on the increments of $q$. Physically one should think of nonlinear springs coupling neighboring displacements with $U$ the corresponding potential and $-U'$ the force. The dynamics is hamiltonian and obtained from

$$H = \sum_{j \in \mathbb{Z}} \left( \tfrac{1}{2} p_j^2 + U(q_{j+1} - q_j) \right), \tag{2}$$

$q_j, p_j$ position and momentum of the $j$-th particle. For the harmonic potential, i.e., $U(x) = \tfrac{1}{2}x^2$, Eq. (1) becomes the standard discrete linear wave equation. For a non-linear force, rather naively one would expect the dynamics to become chaotic. But based on the Kolmogorov-Arnold-Moser (KAM) theorem a mixed phase space, partially chaotic and partially quasi-periodic, is the more likely scenario. On the other hand, the domain wall setting is concerned with the infinite lattice, and how much of the KAM structure persists is a poorly understood subject. We proceed here with a scheme which seems to work in practice.

It will be convenient to first introduce the increments

$$r_j = q_{j+1} - q_j, \tag{3}$$

more physically referred to as stretch, which can have either sign. Clearly, $H$ is the sum of translates of single site terms,

$$H = \sum_{j \in \mathbb{Z}} e_j, \qquad e_j = \tfrac{1}{2} p_j^2 + U(r_j), \tag{4}$$

with $e_j$ the translate of $e_0$ to lattice site $j$. $e_0$ is a smooth function of its arguments. The equations of motion (1) then turn into

$$\dot{r}_j = p_{j+1} - p_j, \qquad \dot{p}_j = U'(r_j) - U'(r_{j-1}). \tag{5}$$

From the dynamics one reads off three local conservation laws, namely stretch, momentum and energy,

$$(r_j, p_j, e_j), \tag{6}$$

and their currents

$$\left( -p_j, -U'(r_{j-1}), -p_j U'(r_{j-1}) \right). \tag{7}$$

While this derivation is straightforward, in principle there could be some other smooth function $h_0$, its translate to site $j$ denoted by $h_j$, which is strictly local in the sense to depend only on finitely many $r_j$'s and $p_j$'s and moreover satisfies the discrete conservation law

$$\frac{d}{dt} h_j = \mathcal{J}_j - \mathcal{J}_{j+1}. \tag{8}$$

If such a current density $\mathcal{J}_j$ exists at all, it has to be smooth and strictly local. As obvious from (8), the time change of $\sum_{j=m}^{j=m'} h_j$ is then only through the boundary of the interval $[m, \ldots, m']$. As an outstanding puzzle in the subject, a strict dichotomy seems to be valid. The nonlinear

wave equation (1) has either three or a countably infinite number of local conservation laws, the former case being called non-integrable. Integrable chains are exceptional. The obvious candidate is the harmonic chain, $U(x) = \frac{1}{2}x^2$, see for example the discussion in [10]. Since the equations of motion are linear, the superposition principle holds. Thus, beyond being integrable the harmonic chain is also non-interacting. The respective GHD is a system of uncoupled conservation laws. The Toda chain, to be discussed below, is also integrable. But now GHD is a nonlinear system of coupled conservation laws, reflecting that conserved fields interact.

To construct an initial domain wall state for a non-integrable chain, note that the dual thermodynamic parameters are $(P, \mathsf{u}, \beta)$, pressure, mean velocity, and inverse temperature. Since $H$ of (4) is a sum of one-point functions, in thermal equilibrium the $\{r_j, p_j\}$'s are independent and identically distributed with single site probability density function

$$\mu_{P,\mathsf{u},\beta}(r_0, p_0) = Z(P, \mathsf{u}, \beta)^{-1} \exp\left[ -\beta\left(\tfrac{1}{2}(p_0 - \mathsf{u})^2 + U(r_0) + Pr_0\right)\right], \qquad (9)$$

$Z$ the normalizing partition function and $P > 0$, $\beta > 0$. The thermal state does not change under the dynamics (5). For a domain wall state we prescribe left and right parameters, $(P_\pm, \mathsf{u}_\pm, \beta_\pm)$. The $(r, p)$'s are still independent, but for $j < 0$ we choose the parameter set $(P_-, \mathsf{u}_-, \beta_-)$ and for $j \geq 0$ the parameter set $(P_+, \mathsf{u}_+, \beta_+)$.

One would expect that on a macroscopic scale the state can be well characterized by block averaged values of the conserved fields. Given these values, the local distribution is close to the corresponding equilibrium state. Assuming such local equilibrium, one arrives at the coupled set of hydrodynamic equations

$$\partial_t \mathfrak{l} + \partial_x \mathsf{j}_{\mathfrak{l}} = 0, \quad \partial_t \mathfrak{u} + \partial_x \mathsf{j}_{\mathfrak{u}} = 0, \quad \partial_t \mathfrak{e} + \partial_x \mathsf{j}_{\mathfrak{e}} = 0, \qquad (10)$$

where the hydrodynamic currents are given by

$$(\mathsf{j}_{\mathfrak{l}}, \mathsf{j}_{\mathfrak{u}}, \mathsf{j}_{\mathfrak{e}}) = \left( -\mathfrak{u}, P(\mathfrak{l}, \mathfrak{e} - \tfrac{1}{2}\mathfrak{u}^2), \mathfrak{u}P(\mathfrak{l}, \mathfrak{e} - \tfrac{1}{2}\mathfrak{u}^2)\right). \qquad (11)$$

Here $\mathfrak{l}(x, t), \mathfrak{u}(x, t), \mathfrak{e}(x, t)$ are the hydrodynamic fields of stretch, velocity, and total energy. $P$ is the pressure depending on stretch and internal energy. From the microscopic model $P$ is obtained by setting $\mathsf{u} = 0$ in Eq. (9) and computing the averages $\langle r_0 \rangle_{P,0,\beta}$, $\langle e_0 \rangle_{P,0,\beta}$. By inverting these functions one arrives at $P$. The Euler equations have to be solved with domain wall initial conditions

$$\left(\mathfrak{l}(x, 0), \mathfrak{u}(x, 0), \mathfrak{e}(x, 0)\right) = \chi(\{x < 0\})\left(\mathfrak{l}_-, \mathfrak{u}_-, \mathfrak{e}_-\right) + \chi(\{x \geq 0\})\left(\mathfrak{l}_+, \mathfrak{u}_+, \mathfrak{e}_+\right). \qquad (12)$$

In our context of a coupled set of hyperbolic conservation laws, (10) – (12) are known mathematically as Riemann problem, which has been thoroughly studied, see [11] for a very readable account. Here, let us merely observe that the solution is self-similar in the sense that it depends on $x, t$ only through the ratio $x/t$. The solution consists of spatial intervals either with constant profiles or smoothly varying profiles, the latter known as rarefaction waves. Possibly they are separated by jump discontinuities, called shocks. To our knowledge there are only a few studies which compare molecular dynamics of the chain with domain wall initial conditions to numerical solutions of the respective hydrodynamic equations. In [12, 13] the program is carried out for a hamiltonian with point hard core potential and alternating masses. Actually in this investigation the goal was to elucidate fluctuations of time-integrated currents at a location inside a rarefaction wave. In fact, the most fascinating part of the study lies in observing how the microscopic motion manages to create and maintain a jump discontinuity. Along the same lines is the recent numerical study of blast formation [14].

The solution to the Riemann problem relies on properties of the Euler equation linearized at local equilibrium, in our case a $3 \times 3$ matrix with matrix elements depending on the thermodynamic parameters. As the location $x$ of the self-similar solution is varied, the corresponding

thermodynamic parameters change and, in order to satisfy the boundary conditions, may be forced to switch between two branches characterized by distinct eigenvalues. In a rough sense, this is the mathematical mechanism behind the formation of shocks. In contrast, for integrable systems the linearization operator has continuous spectrum and no mechanism for shock formation seems to be available.

## 3  GHD of the Toda lattice

The Toda lattice is an anharmonic chain, for which the interaction potential is specified as $U(x) = \mathrm{e}^{-x}$. Hence the hamiltonian is written as

$$H = \sum_{j \in \mathbb{Z}} \left( \tfrac{1}{2} p_j^2 + \mathrm{e}^{-r_j} \right), \quad r_j = q_{j+1} - q_j. \tag{13}$$

In terms of the Flaschka variables [15],

$$a_j = \mathrm{e}^{-r_j/2}, \tag{14}$$

the Lax matrix is the tridiagonal real symmetric matrix with matrix elements

$$L_{j,j} = p_j, \quad L_{j,j+1} = L_{j+1,j} = a_j. \tag{15}$$

For the finite lattice $[1,\dots,N]$ with periodic boundary conditions the $N$ eigenvalues of $L$ are conserved. As functions on phase space they are non-local [16], their local version being $\mathrm{tr}[L^m]$, $m = 1,2,\dots$ [15]. In generalized hydrodynamics such conserved fields are usually called charges or conserved charges and it is convenient to follow this practice. The locally conserved charges of the Toda lattice have a strictly local density given by

$$Q_j^{[m]} = (L^m)_{j,j}, \quad m = 1,2,\dots, \tag{16}$$

with $j \in \mathbb{Z}$. In addition, the stretch is conserved with density

$$Q_j^{[0]} = r_j. \tag{17}$$

The respective current densities are of the form

$$J_j^{[0]} = -Q_j^{[1]}, \qquad J_j^{[m]} = (L^m L^\downarrow)_{j,j}, \tag{18}$$

where $L^\downarrow$ denotes the lower triangular part of $L$, see [17, 21, 22]. By construction one arrives at the microscopic conservation laws

$$\frac{d}{dt} r_j = -p_j + p_{j+1}, \qquad \frac{d}{dt} Q_j^{[m]} = J_j^{[m]} - J_{j+1}^{[m]}, \quad m = 1,2,\dots. \tag{19}$$

Because of the extensive number of conserved charges, thermal equilibrium has to be extended to generalized Gibbs ensembles (GGE). Generalized hydrodynamics is then obtained by averaging (19) in a local GGE state. There is some freedom in the choice of coordinates. The most convenient one comes from the observation that the GGE expectation of $Q^{[m]}$ is the $m$-th moment of the density of states (DOS) of the Lax matrix. Therefore, the natural hydrodynamic fields are the DOS of the Lax matrix and in addition the stretch, $\nu$, both depending on the macroscopic space-time point $(x,t)$. More details can be found in [18–20] and the very recent lecture notes [22]. In standard notation, the DOS is written as $\nu \rho_{\mathrm{p}}(\nu)$, where $\rho_{\mathrm{p}}$ is the

"particle density in spectral parameter space". Then the generalized hydrodynamic equations are

$$\partial_t \nu - \partial_x q_1 = 0, \qquad \partial_t(\nu \rho_\mathrm{p}) + \partial_x\big((\nu^{\mathrm{eff}} - q_1)\rho_\mathrm{p}\big) = 0. \tag{20}$$

Here $q_1 = \nu \int_{\mathbb{R}} \mathrm{d}w\, w \rho_\mathrm{p}(w)$ is the average momentum. The definition of the effective velocity is more lengthy. One introduces the integral operator

$$T\psi(w) = 2\int_{\mathbb{R}} \mathrm{d}w' \log|w - w'|\,\psi(w'), \quad w \in \mathbb{R}, \tag{21}$$

resulting from the two-particle scattering shift of the Toda lattice. Then, for given $\rho_\mathrm{p}$, the effective velocity is the solution of the linear integral equation

$$v^{\mathrm{eff}}(v) = v + \big(T\rho_\mathrm{p}v^{\mathrm{eff}}\big)(v) - \big(T\rho_\mathrm{p}(v)\big)v^{\mathrm{eff}}(v), \tag{22}$$

where $\rho_\mathrm{p}(v)$ is viewed as a multiplication operator.

Writing GHD in this way, the domain wall problem looks inaccessible. Surprisingly, as a general property of GHD, one can transform to normal modes in such a way that the quasi-linear operator is diagonal. This transformation is accomplished by

$$n(v) = \frac{\rho_\mathrm{p}(v)}{1 + \big(T\rho_\mathrm{p}\big)(v)}, \tag{23}$$

and the resulting normal form of the hydrodynamic equations reads

$$v\partial_t n + (v^{\mathrm{eff}} - q_1)\partial_x n = 0, \tag{24}$$

see [21] for a proof. Note that the two-system (20) has merged into a single equation. The symbol $n(v)$ denotes the "number density in spectral parameter space". In [17, 21, 22], the same object is denoted by $\rho_\mu$.

Below we need some further notions from GHD. The dressing transformation of a general function $\psi$ is given by

$$\psi^{\mathrm{dr}} = \psi + Tn\psi^{\mathrm{dr}}, \quad \psi^{\mathrm{dr}} = \big(1 - Tn\big)^{-1}\psi. \tag{25}$$

One also uses the notation $[w^m]^{\mathrm{dr}}$ for the dressing of the $m$-th power of the linear function. Note that the dressing is relative to $n$. After a few algebraic transformations based on (22), (23) and (25), the thermodynamic quantities can be expressed in terms of $n$, which will be convenient for the domain wall problem discussed below. Specifically, one arrives at

$$\rho_\mathrm{p}(v) = n(v)[1]^{\mathrm{dr}}(v). \tag{26}$$

A seemingly more explicit formula for the effective velocity is

$$v^{\mathrm{eff}}(v) = \frac{[w]^{\mathrm{dr}}(v)}{[1]^{\mathrm{dr}}(v)}. \tag{27}$$

In addition

$$v^{-1} = \int_{\mathbb{R}} \mathrm{d}w\, \rho_\mathrm{p}(w) = \int_{\mathbb{R}} \mathrm{d}w\, n(w)[1]^{\mathrm{dr}}(w) \qquad \text{and} \qquad q_1 = v\int_{\mathbb{R}} \mathrm{d}w\, n(w)[w]^{\mathrm{dr}}(w). \tag{28}$$

*Thermal states.* — As discussed in [17, 19, 22, 23], for thermal states, $n(v)$ is a solution of the thermodynamic Bethe ansatz (TBA) equation,

$$V(w) - \mu - (Tn)(w) + \log n(w) = 0, \tag{29}$$

with $V(w) = \frac{1}{2}\beta w^2$. The "chemical potential" $\mu$ is a Lagrange multiplier which ensures normalization as

$$\int_{\mathbb{R}} \mathrm{d}w\, n(w) = P\,. \tag{30}$$

$\mu$ depends on the thermodynamic parameters $\beta$ and $P$, and turns out to be equal to the free energy $F(\beta, P)$, for which an explicit formula is available [17, 22],

$$F(\beta, P) = \log\sqrt{\beta/(2\pi)} + P\log\beta - \log\Gamma(P)\,, \tag{31}$$

with Gamma function, $\Gamma$. The thermally averaged stretch is then the derivative of the free energy with respect to the pressure,

$$\nu = \partial_P F(\beta, P) = \log\beta - \psi(P)\,, \tag{32}$$

$\psi$ denoting the digamma function.

Fig. 1 visualizes these thermal functions. $F(\beta, P)$ is a concave function, and assumes its maximum at the critical pressure $P_c > 0$. Thus the average stretch $\nu$ decreases with increasing $P$ and crosses zero at $P_c$. For low pressure particles are far apart and the density is small. As $P$ is increased, the free volume between particles with neighboring index shrinks and vanishes at $P_c$. Upon further increase the free volume turns negative. From the shape of $F(\beta, P)$ (and thus of $\mu$) one concludes that for given $\mu$ there are two different values of $P$. In (29) only $\mu$ appears as parameter. Thus this equation has two solution branches, low and high pressure, which match at $P_c$. As $P$ is moved through $P_c$, $\nu\rho_p$ and $n$ vary smoothly. On the other hand, $\rho_p$ has to diverge as $P \to P_c$, since $\nu = 0$ at $P_c$. Numerically we investigated only thermal equilibrium. But any other GGE is expected to show the same behavior.

*Numerical method.* — We first discuss the computation of the thermal $n(w)$: TBA is solved numerically via Newton iteration (Mathematica's FINDROOT), together with the quasi-energy parameterization $n(w) = \mathrm{e}^{-\varepsilon(w)}$ to ensure $n(w) > 0$. In practice, this requires a good starting point, which we obtain via an associated Fokker-Planck equation [23]. As sketch of a derivation, first insert $n(\nu) = P\rho_s(\nu)$ into (29), which yields

$$V(w) - \mu - P(T\rho_s)(w) + \log\rho_s(w) + \log P = 0\,. \tag{33}$$

Differentiating this equation with respect to $w$ and then multiplying by $\rho_s(w)$ results in

$$V'(w)\rho_s(w) - 2P\int_{\mathbb{R}} \mathrm{d}\nu\, \frac{1}{w - \nu}\rho_s(\nu)\rho_s(w) + \rho_s'(w) = 0\,, \tag{34}$$

$\rho_s$ can be interpreted as the stationary solution of the time-dependent nonlinear Fokker-Planck equation

$$\partial_t\rho(w, t) = \partial_w\left(V'(w)\rho(w, t) - 2P\int_{\mathbb{R}} \mathrm{d}\nu\, \frac{1}{w - \nu}\rho(\nu, t)\rho(w, t)\right) + \partial_w^2\rho(w, t)\,. \tag{35}$$

Requiring the normalization $\int_{\mathbb{R}} \mathrm{d}w\rho_s(w) = 1$, Eq. (35) has a unique stationary solution. While TBA and Fokker-Planck are equivalent analytically, we observed that, numerically, optimal precision is obtained by first solving the nonlinear Fokker-Planck equation, and then using these data as initial value for the TBA Newton iteration.

Regarding GHD, the key task is to realize the dressing transformation (25) numerically. According to (25), this amounts to solving a linear system of equations after discretization. We have found a finite element discretization of $n(\nu)$, $\rho_p(\nu)$, $v^{\mathrm{eff}}(\nu)$, etc. via *hat functions* on a uniform grid to work well in practice. Specifically, we use the grid spacing $h = \frac{1}{10}$. The action

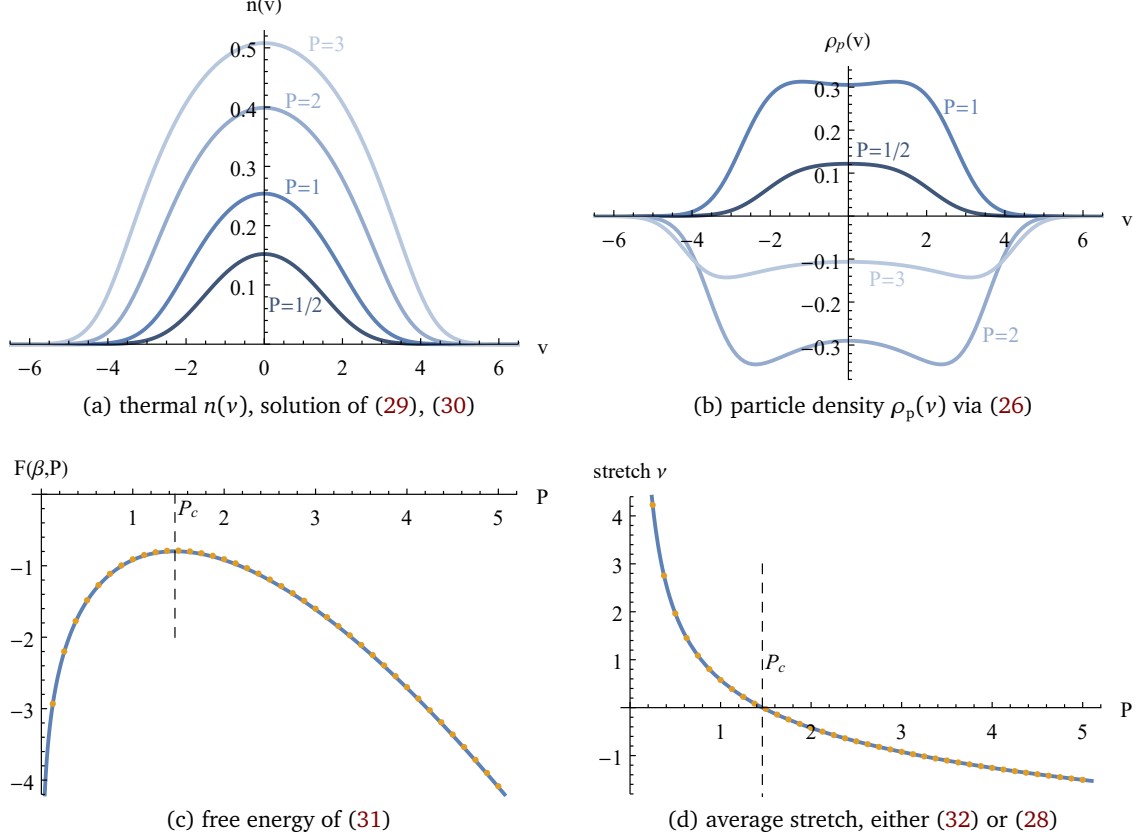

(a) thermal $n(v)$, solution of (29), (30)

(b) particle density $\rho_p(v)$ via (26)

(c) free energy of (31)

(d) average stretch, either (32) or (28)

Figure 1: Thermal states for inverse temperature $\beta = 1$. (a) The number density $n(v)$ as obtained by solving the TBA equation (29) with normalization (30), for different values of $P$. (b) The corresponding particle density $\rho_p(v)$ diverges at $P_c$ and flips its overall sign as $P$ crosses $P_c$. (c) Free energy (31), which coincides with the chemical potential $\mu$ appearing in (29). As consistency check, the yellow dots show $\mu$ obtained by inserting the Fokker-Planck stationary solution, (35), scaled by $P$ into (29). (d) Average stretch: the solid line shows (32) and yellow dots the stretch computed via (28).

of the integral operator (21) can be symbolically precomputed for these hat functions, such that the discretization of $T$ is a symmetric matrix.

Starting from a given $n(v)$, one subsequently calculates $[1]^{\text{dr}}(v)$ and $[w]^{\text{dr}}(v)$, then $\rho_p(v)$ via (26) and $v^{\text{eff}}(v)$ via (27), as well as $v$ and $q_1$ via (28). Due to the discretization on a uniform grid, the integration in (28) amounts to a simple summation of the hat function coefficients of the integrand.

Regarding the domain wall problem discussed in the following section, $n^\phi(v)$ turns out to be a piecewise combination of thermal functions, see Eq. (39) below. For such a $n^\phi(v)$, we use the algorithm just described to obtain $[1]^{\text{dr}}(v)$, $\rho_p(v)$, etc..

A Mathematica implementation of the numerical method and the simulations used for this work are available at [31].

# 4 Solution to the domain wall initial condition

We slightly rewrite Eq. (24) as

$$\partial_t n(x,t;v) + \tilde{v}^{\mathrm{eff}}(x,t;v)\partial_x n(x,t;v) = 0, \quad \tilde{v}^{\mathrm{eff}}(v) = v^{-1}(v^{\mathrm{eff}}(v) - q_1). \tag{36}$$

The domain wall initial conditions are

$$n(x,0;v) = \chi(\{x < 0\})n_-(v) + \chi(\{x \geq 0\})n_+(v). \tag{37}$$

Instead of $n_\pm$, physically it might be more natural to prescribe the DOS of the Lax matrix and the average stretch. But mathematically the normal form (36) is more accessible.

Since the solution to (36), (37) scales ballisticly, we set $n(x,t;v) = \mathsf{n}(t^{-1}x;v)$ and $\tilde{v}^{\mathrm{eff}}(x,t;v) = \tilde{\mathsf{v}}^{\mathrm{eff}}(t^{-1}x;v)$. Without loss of generality one adopts $t = 1$ and arrives at

$$(x - \tilde{\mathsf{v}}^{\mathrm{eff}}(x;v))\partial_x \mathsf{n}(x;v) = 0, \qquad \lim_{x \to \pm\infty} \mathsf{n}(x;v) = n_\pm(v). \tag{38}$$

Therefore $x \mapsto \mathsf{n}(x;v)$ for fixed $v$ has to be constant except for jumps at the zeros of $x \mapsto x - \tilde{\mathsf{v}}^{\mathrm{eff}}(x;v)$. At this stage, it is not so clear at which level of generality to proceed. Physically, one would expect to have a unique solution. Thus in the $(x,v)$-plane there should be a *contact line* which divides the plane into two domains, one containing the set $\{-\infty\} \times \mathbb{R}$ and the other the set $\{\infty\} \times \mathbb{R}$. In addition, the contact line has the property to be the graph of some function $v \mapsto \tilde{\phi}(v)$. The solution $\mathsf{n}(x;v)$ equals $n_\pm(v)$ on either domain with a jump across the contact line. In the example of the harmonic chain, the parameter $v$ corresponds to the wave number which varies only over a bounded interval. In this case the contact line function cannot be inverted. On the other hand, for hard rods [6], $v$ varies over the entire real axis and the contact line is monotone increasing. Since the Toda lattice is closer to hard rods, we will assume that the contact line is given by the inverse of $\tilde{\phi}$, denoted then by $\phi$.

With such an assumption, for every $x$ there is a contact point $\phi(x)$. Then the solution ansatz reads

$$n^\phi(v) = \chi(\{v > \phi\})n_-(v) + \chi(\{v \leq \phi\})n_+(v) \text{ and } \mathsf{n}(x;v) = n^{\phi(x)}(v). \tag{39}$$

The superscript $\phi$ will be used to generically indicate that in the TBA formalism $n^\phi$ is substituted for $n$ and similarly the subscript $\pm$ signals the substitution of $n_\pm$. For example, compare with (25), (28),

$$\psi^{\mathrm{dr},\phi} = \left(1 - Tn^\phi\right)^{-1}\psi, \quad (v_\pm)^{-1} = \int_{\mathbb{R}} dw \rho_{\mathrm{p}\pm}(w) = \int_{\mathbb{R}} dw\, n_\pm(w)[1]^{\mathrm{dr},\pm}(w). \tag{40}$$

We set

$$\tilde{\mathsf{v}}^{\mathrm{eff}}(x;v) = \tilde{v}^{\mathrm{eff},\phi(x)}(v). \tag{41}$$

Then the condition on the left side of (38) translates to

$$x = \tilde{v}^{\mathrm{eff}}(x, \phi(x)). \tag{42}$$

Put differently, one defines

$$G(\phi) = \tilde{v}^{\mathrm{eff},\phi}(\phi), \tag{43}$$

then

$$x = G(\phi(x)), \qquad \phi(x) = G^{-1}(x), \tag{44}$$

which means that $G$ is the inverse of the contact line $\phi$.



(a) $n^\phi(v)$ defined in (39), for $\phi = \frac{3}{2}$

(b) Lax density of states $v^\phi \rho_p^\phi(v)$, for $\phi = \frac{3}{2}$

(c) stretch $v^\phi$ obtained by (28)

(d) average momentum via (28)

(e) $\tilde{v}^{\mathrm{eff},\phi}(v)$ in (36), (41) for $\phi = 2$

(f) corresponding $G(\phi)$ in (43)

Figure 2: Numerical domain wall simulation results. The initial thermal states in the left and right domain (dashed curves) are specified by inverse temperature $\beta = 1$ and pressure $P_- = \frac{1}{2}$, $P_+ = 2$, respectively. Even though $v^\phi$ crosses zero around $\phi = 2$, the rescaled effective velocity $\tilde{v}^{\mathrm{eff},\phi}$ remains finite.

While $G(\phi)$ itself has to be obtained numerically, the large $|\phi|$ asymptotics can still be argued analytically. We start from

$$v^{\mathrm{eff},\phi}(\phi)\bigl(1 + T\rho_{\mathrm{p}}^{\phi}(\phi)\bigr) = \phi + \bigl(T\rho_{\mathrm{p}}^{\phi} v^{\mathrm{eff},\phi}\bigr)(\phi). \tag{45}$$

Setting

$$n^{\phi}(v) = n_{+}(v) + \chi(\{v > \phi\})(n_{-}(v) - n_{+}(v)), \tag{46}$$

we note that for large $\phi$ the second term is exponentially small and can be neglected. The second summand on the left of (45) then reads

$$2\int_{\mathbb{R}} \mathrm{d}w \log|\phi - w|\rho_{\mathrm{p},+}(w) \simeq 2\log\phi \int_{\mathbb{R}} \mathrm{d}w \rho_{\mathrm{p},+}(w) = 2(v_{+})^{-1}\log\phi. \tag{47}$$

Similarly for the right side of (45) one finds a logarithmic increase with a prefactor $c_{+}$, which could be determined by a further iteration. For $\phi \to \infty$ the term $q_{1}^{\phi}/v^{\phi}$ converges to $q_{1,+}/v_{+}$. Thus one arrives at the large $\phi$ asymptotics,

$$G(\phi) \simeq \frac{\phi + c_{+}\log|\phi|}{v_{+} + 2\log|\phi|} - \frac{q_{1,+}}{v_{+}}. \tag{48}$$

For $\phi \to -\infty$, the same asymptotics holds upon substituting $v_{-}, c_{-}$ for $v_{+}, c_{+}$. If the boundary functions $n_{\pm}$ have exponential decay, the error in (48) would be of the same order.

*Exemplary numerical solution.* — The considerations above hold for an arbitrary choice of boundary values $n_{\pm}$. For the simulation we impose thermal boundaries, characterized by zero average momentum, constant inverse temperature $\beta = 1$, and pressures $P_{-} = \frac{1}{2}$, $P_{+} = 2$, respectively, such that $P_{-} < P_{\mathrm{c}} < P_{+}$. The results are shown in Fig. 2. Starting from $n^{\phi}(v)$ defined in (39), all the other hydrodynamic functions can be obtained via dressing transformations. While $n^{\phi}(v)$ pointwise agrees with $n_{\pm}(v)$ by construction, see Fig. 2a, this is no longer the case for the normalized particle density $v^{\phi}\rho_{\mathrm{p}}^{\phi}(v)$ shown in Fig. 2b. For comparison, the dashed curves in Fig. 2 visualize the functions computed for thermal $n_{\pm}(v)$.

Only a single low-high pressure domain wall has been simulated in detail. We tested a few other values. As physically to be expected, the resulting plots look rather similar. Only the condition $P_{-} < P_{\mathrm{c}} < P_{+}$ has to be respected.

## 5  Toda fluid and related work

So far we viewed the Toda lattice as a discrete nonlinear wave equation. A physically more immediate perspective would be to have particles moving on the real line, which is referred to as Toda fluid. Instead of stretch, one then considers the fluid density $\rho_{\mathrm{f}}$. For a homogeneous system $|v|\rho_{\mathrm{f}} = 1$. Including space-time variations, the mapping between lattice and fluid is discussed in [18]. Rather unexpected features appear for the low-high pressure domain wall, however. We consider the initial state and set $q_{0} = 0$. Then $\{q_{j+1} - q_{j}, j \leq -1\}$ are i.i.d. random variables and so are $\{q_{j+1} - q_{j}, j \geq 0\}$. By assumption $\langle q_{j+1} - q_{j}\rangle = v_{-} > 0$ for $j \leq -1$, while $\langle q_{j+1} - q_{j}\rangle = v_{+} < 0$ for $j \geq 0$. A typical ordering is of the form $\cdots < q_{-1} < q_{0} = 0 > q_{1} > \ldots$. Thus initially the average particle density equals $v_{-} + |v_{+}|$ on $(-\infty, 0]$ and decays exponentially on $[0, \infty)$. Under the dynamics, the point at which the ordered domain touches the anti-ordered domain is moving in time, its label being denoted by $\kappa(t)$. Close to $q_{\kappa(t)}$ particles pile up. The domain boundary acts as bottleneck for particles. The position of the bottleneck is $x_{\mathrm{bn}}(t) = q_{\kappa(t)}$. Our numerical simulations suggest that $\kappa(t)$ and $x_{\mathrm{bn}}(t)$ change linearly in time, at least approximately. Also beyond the bottleneck position the particle density vanishes

rapidly. As observed numerically, the scaling function $g$ has a nonzero slope at $x_c$. Thus, near the bottleneck the particles should be distributed according to equilibrium with a linearly varying pressure. From this property one infers that in physical space the particle density at the bottleneck diverges as an inverse square root. It would be of interest to better understand the precise particle statistics close to the bottleneck.

Within GHD, domain wall initial conditions have been studied numerically also for a discrete version of the sinh-Gordon model [26]. Most detailed studies are available for the XXZ model [27, 28]. In this case the spectral parameter space has in addition the type of string states and the contact line carries extra indices. Because of momentum conservation, the contact line of the Toda lattice is supported on the full real line. For XXZ, and other discrete models, one usually observes a light cone, i.e., the contact line is supported on an interval and the boundary values are maintained up to an edge. The behavior near the edge often shows then an intricate oscillatory decay, which has been elucidated in considerable detail [29, 30].

On the Euler scale the self-similar solution has a sharp step at the contact line. One expects that, because of dissipation, the step is actually broadened to an error-like function. For the Toda lattice, and in general for integrable systems, the structure of diffusive corrections are available [24]. But respective numerical simulations for the Toda lattice are still missing. On the other hand for the XXZ chain, both on the microscopic and GHD level, broadening has been convincingly observed [25].

# Acknowledgements

CM acknowledges support from the Munich Center for Quantum Science and Technology. HS thanks Tomohiro Sasamoto for his generous hospitality at Tokyo Institute of Technology, where the initial work was accomplished, and Benjamin Doyon for teaching on domain walls.

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
