# Peer review of "High-low pressure domain wall for the classical Toda lattice"

_SciPost Physics, doi:SciPost Phys. Core 5, 002 (2022)_

## Round 1 · Referee Report · Nicolas Nessi (Referee 1) · 2021-2-15

Strengths

1- The paper is well written and presents very well the context of the problem which is analyzed. 2- The subject of the paper is timely and adds new results to a field which is growing, namely, the non-equilibrium dynamics of isolated integrable models, and generalized hydrodynamics in particular. 3- The paper includes a pedagogical introduction to the domain wall problem in section 2.

Weaknesses

1- At some points the exposition seems too compressed. Adding more details could help to make the exposition clearer to a broader public.

Report

The authors study the domain wall problem in the Toda chain using generalized hydrodynamics. The most important result is that, for some special parameter values specifying the initial condition, a singularity in the particle density develops a singularity for finite times.
I consider that the paper meets the publication criteria of the journal and recomend publication with minor revisions.

Requested changes

1- Before equation 2.13 it might be useful to explicitly define the linear operator $W$. 2- In order to render the presentation more self-contained and for the sake of clarity, before equation 3.8, provide more details about the GHD equations. 3- The singularity in the non-equilibrium behavior seems to be related to the existence of a critical pressure in equilibrium. Could you elaborate on this relationship? 4- Is it possible to specify the parameter region for which this singularity appears?Is there any identifiable condition on $P_+$ and $P_{-}$ that ensures the appearence of the singularity? As far as I understand, the authors have studied only one point for which the singularity is present.

---

## Round 1 · Referee Report · Anonymous (Referee 2) · 2021-3-1

Strengths

Interesting new phenomenon reported

Weaknesses

Clarity of exposition

Report

The paper studies the dynamics in classical Toda lattice from initial domain wall initial conditions.
The framework used for the analysis is that of Generalized Hydrodynamics (GHD).

The focus is on a particular phenomenon observed for a certain class of initial states (thermal states with different temperature, mean velocity and temperature, for specific range of parameter). The phenomenon seems specific to the model under consideration.

While the content is interesting, I think that the clarity of the exposition can be definitely improved.
In "requested changes" I report some example of concepts to be better explained, in order for the paper to be more self-contained and understandable.
After such revision is made, I recommend it for publication is Scipost.

Requested changes

  • What is the self-similar solution of GHD? Something is said after eq. 2.14, but should be clearly stated in the paper.
  • What is W in eq. (2.13)? I guess it is the Wigner function, but it is not defined
  • As far as I understand the paragraph “numerical method” (pag 8) is the algorithm for the equilibrium problem (the initial condition) and not for the GHD equation: I would specify it, since the section is called “GHD of the Toda lattice”
  • Simulations with molecular dynamics/hard rods are mentioned more than once, but not properly introduced

---

## Round 1 · Referee Report · Anonymous (Referee 3) · 2021-4-12

Strengths

  • Pedagogical introduction to the generalized hydrodynamics and tba description of Toda lattices

Weaknesses

  • In some parts, the notation is heavy and hard to follow
  • No heuristic mechanism for the development of the density singularity is discussed
  • Lack of ab initio numerical simulation to be compared with the GHD

Report

The authors study transport in the classical Toda lattice, focusing on the famous partitioning protocol where two infinite halves of the system are initialized in two different states (in this case, thermal states) and then let evolve.
The Toda lattice is integrable, hence the authors approach the problem within the framework of generalized hydrodynamics (GHD) and report on a singularity in the profiles of the local particle density, which is their main contribution.
I believe the results are interesting and worth to be communicated, but I think they best appeal to a more specialistic audience, hence ScipostCore is probably more appropriate.
In general, I appreciate the pedagogic and detailed introduction the authors provide to the concepts of GHD and in particular of the Toda lattice, but I think in some parts the manuscript is hard to follow and I would ask my comments are taken into account before suggesting the paper for publication.
I feel that not enough space is given to discussing the singularity (which is the central result of the paper) and appears only at the very end of Section 4. Besides, I have some further technical questions that I ask the authors to address.

1. Is there any physical picture or interpretation that can explain such a singularity? If it is so, I think it would important to explicitly mention it.

2. Which are the conditions for the development of such a singularity? Why was it not observed, for example, in Ref. [19]?

3. Since the natural output of the GHD equations is n (which is linked to rho through Eq. (19)), and n is bounded, I do not see why \nu^{-1} diverges. Can the authors comment more on how the singularity is developed in the equations? Is this feature due to the dressing?

4. The singularity 1/|x| is not integrable, hence the number of particles contained in an interval encompassing such a singularity is infinite. However, on the initial state the particle density is finite, hence the singularity is developed through time evolution. How does it happen? Can the authors maybe provide a hydrodynamic description for it? I think that microscopic simulations of the classical model could further elucidate this point, but I understand it requires some extra work.

5. In the conclusions, the authors comment that this singularity seems to be special for the Toda lattice. Can the authors speculate why is it so? What does make Toda special compared with other models? I understand this could be a difficult question, but if the authors have some further insight it would be important to share it.

6. In Eq. (3.8) rho and nu appear for the first time and the authors refer to the preexisting literature for their definition, but maybe some extra comments could make the paper more self-contained. For example, the authors explicitly mention rho to be the particle density, but do not say anything about nu, which is defined only later in Eq. (3.16). Also, if \nu^{-1} is defined as the integral of the particle density, I would expect it to be a positive quantity, while in Fig. 3.c \nu passes through zero and changes sign. Could it be there is a typo in Eq. (3.16) and there is an absolute value missing? I think the absolute value is needed to be consistent with Ref. [17]. Moreover, I would ask the authors to provide an explicit definition of \nu in their manuscript.

7. Lastly, in the abstract it is written “...with a different choice of coordinates smooth behavior is recovered.” and then, at the end of section “Of course, the density really diverges, but when viewed in a different coordinate system the behavior is smooth.” Both sentences sound a bit obscure: which is the coordinate system the authors are referring to? Can they further elaborate on this?

---

## Round 3 · Referee Report · Anonymous (Referee 3) · 2021-11-1

Report

I have read the resubmitted version of the manuscript, as well as the reports of the other referees.
I think all of us pointed out (to some extent) the lack of clarity of the previous version of the submission: the authors greatly improved this point in the current version, which is now much more accessible.
All the points I addressed in my previous report have been thoroughly addressed and clarified in the revised manuscript.
I believe the results are solid and of interest to the community working on integrable models out of equilibrium and in particular their hydrodynamics and I recommend publication.
Yet, I think the submission targets a specialistic audience and it is probably more suited for SciPost Physics Core.

---

## Round 3 · Referee Report · Anonymous (Referee 2) · 2021-11-2

Report

The authors addressed my doubts, implemented the required changes and improved the clarity of the manuscript. I therefore can recommend it for publication.

---

## Round 3 · Referee Report · Nicolas Nessi (Referee 1) · 2021-11-11

Report

The authors have addressed all the points I raised in the first report. Overall, I think that now the physics of the main result is more clearly stated and the paper is more readable. Consequently, I can recommend for publication.

---

## Round 3 · Referee Report · Anonymous (Referee 4) · 2021-11-19

Strengths

  1. Exact results on an interesting interacting classical model of many-body dynamics
  2. The paper is well written and clear, at least for readers with previous experience of GHD/TBA equations

Weaknesses

  1. No comparison with numerical simulations of the microscopic model
  2. The results are rather incremental: this is one paper in a sequence of many other papers on TBA/GHD in the classical Toda lattice

Report

The paper presents results on the domain-wall problem (or Riemann problem) in the hydrodynamic equations that emerge from the Toda lattice at large scale. The problem seems non-trivial because of an apparent singularity in one of the quantities appearing in the thermodynamic description (the particle density $\rho_{\rm p}$), but the authors demonstrate that the physical solution to the hydrodynamic equations remains smooth and that this singularity is not a problem.

I find the paper very interesting, however it is disappointing that no comparison is made with simulations of the microscopic model. It seems that such simulations could be easily doable in the classical Toda lattice, and it would be really nice to see how the microscopic simulation compares to the hydrodynamic solution. Also, it would probably be very illuminating to have illustrations of typical configurations of the Toda lattice in its initial domain-wall state and at later times, in addition to the plots of the hydrodynamic solution that are currently shown in the manuscript.

Otherwise, the paper clearly uses a lot of results on the hydrodynamics of the Toda lattice developed by the authors in the past years. As far as I can see, it is really a sequence of these previous works. Both the Toda lattice and the Riemann problem in GHD have been studied extensively by the authors and others since 2016. What is new here is to combine the two topics: the Riemann problem and the Toda lattice.
There is no doubt that the manuscript deserves to be published, but I am not convinced that the criteria for publication in Scipost Physics are met (these criteria are supposed to be highly selective ). May I suggest to the Editor that the paper could possibly be better suited for Scipost Physics Core?

Requested changes

The manuscript can be published as it is (in Scipost Core/Scipost Physics, see above), it is clear and contains interesting new results. A suggestion for improving the paper, that the authors may or may not follow, is to compare with numerics in the microscopic model.

---

## Round 3 · Author Response

Dear Editor and Reviewers,

We thank you for critically reading our manuscript, and greatly appreciate your helpful comments and suggestions. We have addressed them as follows:

Response to reviewer 1: 1. Before equation 2.13 it might be useful to explicitly define the linear operator $W$. Such a detailed discussion of the non-interacting case has been removed. 2. In order to render the presentation more self-contained and for the sake of clarity, before equation 3.8, provide more details about the GHD equations. The Toda GHD is discussed at length in the recent literature. While trying to avoid duplication, we still added a few more explanations for better readability. 3. The singularity in the non-equilibrium behavior seems to be related to the existence of a critical pressure in equilibrium. Could you elaborate on this relationship? To more clearly state the main result of our contribution, we changed the title, rewrote the introduction, and correspondingly modified the last section. The critical pressure corresponds to a vanishing free volume (= stretch). 4. Is it possible to specify the parameter region for which this singularity appears? Is there any identifiable condition on $P_+$ and $P_-$ that ensures the appearance of the singularity? As far as I understand, the authors have studied only one point for which the singularity is present. The specific values of $P_+$ and $P_-$ are of no importance as long as $P_- < P_\mathrm{c} < P_+$. We have run more simulations for other parameters satisfying the condition. No qualitative change is observed.

Response to reviewer 2: - What is the self-similar solution of GHD? Something is said after eq. 2.14, but should be clearly stated in the paper. What is W in eq. (2.13)? I guess it is the Wigner function, but it is not defined. We now define explicitly the notion of self-similarity. The detailed discussion on the non-interacting case has been removed. So $W$ and Eqs. (2.13), (2.14) no longer appear. - As far as I understand the paragraph numerical method'' (page 8) is the algorithm for the equilibrium problem (the initial condition) and not for the GHD equation: I would specify it, since the section is calledGHD of the Toda lattice''. This is indeed a somewhat misleading formulation. We have expanded and improved our presentation, and also explicitly cover the calculations for GHD. In particular, we emphasize that starting from $n(v)$ allows to compute all the remaining functions of interest. For domain wall only the right/left asymptotics is thermal equilibrium, and the intermediate $n(v)$ has a pointwise jump, as illustrated in Fig. 2(a). - Simulations with molecular dynamics/hard rods are mentioned more than once, but not properly introduced. Our paper does not report on molecular dynamics. But it would be interesting to compare GHD predictions with molecular dynamics. We improved our explanations.

Response to reviewer 3: 1. Is there any physical picture or interpretation that can explain such a singularity? If it is so, I think it would important to explicitly mention it. To more clearly state the main result of our contribution, we changed the title, rewrote the introduction, and correspondingly modified last section. The singularity is linked to the vanishing of the free volume (= stretch). The physical picture of a bottleneck separating low and high pressure regime has been added in Section 5. 2. Which are the conditions for the development of such a singularity? Why was it not observed, for example, in Ref. [19]? The condition is to have a low-high pressure domain wall. In [19] only the case of low-low pressure domain wall is studied. 3. Since the natural output of the GHD equations is $n$ (which is linked to $\rho$ through Eq. (19)), and n is bounded, I do not see why $\nu^{-1}$ diverges. Can the authors comment more on how the singularity is developed in the equations? Is this feature due to the dressing? $\nu^{-1}$ diverges at some intermediate spatial location because the free volume vanishes. We hope that this feature is better explained in the revised version. 4. The singularity $1/|x|$ is not integrable, hence the number of particles contained in an interval encompassing such a singularity is infinite. However, on the initial state the particle density is finite, hence the singularity is developed through time evolution. How does it happen? Can the authors maybe provide a hydrodynamic description for it? I think that microscopic simulations of the classical model could further elucidate this point, but I understand it requires some extra work. Agreed. We reinvestigated the issue. In fact the physical density of particles has an inverse square root divergence, which is integrable. A molecular dynamics simulation is on the agenda. 5. In the conclusions, the authors comment that this singularity seems to be special for the Toda lattice. Can the authors speculate why is it so? What does make Toda special compared with other models? I understand this could be a difficult question, but if the authors have some further insight it would be important to share it. Good point. This is discussed now in Section 5. 6. In Eq. (3.8) $\rho$ and $\nu$ appear for the first time and the authors refer to the preexisting literature for their definition, but maybe some extra comments could make the paper more self-contained. For example, the authors explicitly mention $\rho$ to be the particle density, but do not say anything about $\nu$, which is defined only later in Eq. (3.16). Also, if $\nu^{-1}$ is defined as the integral of the particle density, I would expect it to be a positive quantity, while in Fig. 3.c $\nu$ passes through zero and changes sign. Could it be there is a typo in Eq. (3.16) and there is an absolute value missing? I think the absolute value is needed to be consistent with Ref. [17]. Moreover, I would ask the authors to provide an explicit definition of $\nu$ in their manuscript. We improved this part and added more explanations. In our terminology $\nu$ is the free volume (stretch). By definition, $\nu > 0$ in the low pressure regime and $\nu < 0$ in the high pressure regime. There is no typo in (3.16). $\nu\rho_\mathrm{p}\geq 0$ because it is the DOS of the Lax matrix. When $\nu$ changes from positive to negative values, $\rho_\mathrm{p}$ globally flips its sign, see Fig. 2 (b) in the current version. $\nu$ is now explicitly defined. The common notion of particle density refers to the spectral parameter or root density. This is distinct from the physical density of Toda particles. We tried to make this distinction clear. 7. Lastly, in the abstract it is written ...with a different choice of coordinates smooth behavior is recovered.'' and then, at the end of sectionOf course, the density really diverges, but when viewed in a different coordinate system the behavior is smooth.'' Both sentences sound a bit obscure: which is the coordinate system the authors are referring to? Can they further elaborate on this? Unfortunately these sentences are not so helpful and have been removed. The particle (fluid) picture is now explained in the new Section 5. The low and high pressure particles sit on top of each other in physical space and only cover a half-line. At the end-point of the half-line there is a ``bottleneck" at which the (positive) physical particle density diverges as an inverse square root.

With these changes, we hope that the revised version can be accepted for publication.

Best regards, Christian Mendl and Herbert Spohn

---

## Round 3 · List of Changes

• changed the title to more clearly state the main result
  • omitted Fig. 1 in the original manuscript and 1/|x| singularity discussion
  • defined explicitly the notion of self-similarity in the introduction; added explanations of Toda GHD; removed detailed discussion of the non-interacting case, such that $W$ and Eqs. (2.13), (2.14) no longer appear
  • extended the description of the numerical method, in particular the GHD simulations
  • slightly revised the description and discussion of the domain wall problem in section 4
  • included comparison of Toda with other models in revised concluding section 5
  • physical picture of a bottleneck separating low and high pressure regime added in section 5

---

## Editorial Decision

published